# Carboplatin Dosing in Children Using Estimated Glomerular Filtration Rate: Equation Matters

**DOI:** 10.3390/cancers13235963

**Published:** 2021-11-26

**Authors:** Mirjam E. van de Velde, Emil den Bakker, Hester N. Blufpand, Gertjan L. Kaspers, Floor C. H. Abbink, Arjenne W. A. Kors, Abraham J. Wilhelm, Richard J. Honeywell, Godefridus J. Peters, Birgit Stoffel-Wagner, Laurien M. Buffart, Arend Bökenkamp

**Affiliations:** 1Emma Children’s Hospital, Amsterdam UMC, Vrije Universiteit Amsterdam, Pediatric Oncology, 1081 HV Amsterdam, The Netherlands; h.blufpand@amsterdamumc.nl (H.N.B.); w.a.kors@prinsesmaximacentrum.nl (A.W.A.K.); gjl.kaspers@prinsesmaximacentrum.nl (G.L.K.); 2Department of Pediatric Oncology/Hematology, Amsterdam UMC, 1081 HV Amsterdam, The Netherlands; 3Emma Children’s Hospital, Amsterdam UMC, Vrije Universiteit Amsterdam, Pediatric Nephrology, 1081 HV Amsterdam, The Netherlands; emil.denbakker@amsterdamumc.nl (E.d.B.); a.bokenkamp@amsterdamumc.nl (A.B.); 4Princess Máxima Center for Pediatric Oncology, 3584 CS Utrecht, The Netherlands; 5Emma Children’s Hospital, Amsterdam UMC, Amsterdam Medical Center, Pediatric Oncology, 1081 HV Amsterdam, The Netherlands; f.abbink@amsterdamumc.nl; 6Amsterdam UMC, Vrije Universiteit Amsterdam, Clinical Pharmacology and Pharmacy, 1081 HV Amsterdam, The Netherlands; aj.wilhelm@amsterdamumc.nl; 7Laboratory of Medical Oncology, Amsterdam University Medical Center, VUMC, 1081 HV Amsterdam, The Netherlands; r.honeywell@amsterdamumc.nl (R.J.H.); gj.peters@amsterdamumc.nl (G.J.P.); 8Department of Biochemistry, Medical University of Gdansk, 80-210 Gdansk, Poland; 9Institute for Clinical Chemistry and Clinical Pharmacology, University of Bonn-Medical Center, 53127 Bonn, Germany; Birgit.Stoffel-Wagner@ukbonn.de; 10Department of Epidemiology and Biostatistics, Amsterdam UMC, Vrije Universiteit Amsterdam, 1081 HV Amsterdam, The Netherlands; l.buffart@amsterdamumc.nl

**Keywords:** carboplatin, children, renal function-based dosing, glomerular filtration rate, cystatin C, retinoblastoma

## Abstract

**Simple Summary:**

Carboplatin is a chemotherapeutic agent that is usually dosed based on body surface area or weight. However, carboplatin is cleared from the body by the kidneys. Therefore, taking the patient’s kidney function into account to calculate the adequate dose of carboplatin might result in a better exposure of carboplatin within a patient. In this study we sought to validate a carboplatin dosing method based on kidney function and compare several methods for kidney function-based carboplatin dosing in children suffering from retinoblastoma. We were able to show that carboplatin dosing based on a marker of kidney function (cystatin C) resulted in more adequate dosing than dosing on body surface area or weight.

**Abstract:**

Renal function-based carboplatin dosing using measured glomerular filtration rate (GFR) results in more consistent drug exposure than anthropometric dosing. We aimed to validate the Newell dosing equation using estimated GFR (eGFR) and study which equation most accurately predicts carboplatin clearance in children with retinoblastoma. In 13 children with retinoblastoma 38 carboplatin clearance values were obtained from individual fits using MWPharm++. Carboplatin exposure (AUC) was calculated from administered dose and observed carboplatin clearance and compared to predicted AUC calculated with a carboplatin dosing equation (Newell) using different GFR estimates. Different dosing regimens were compared in terms of accuracy, bias and precision. All patients had normal eGFR. Carboplatin exposure using cystatin C-based eGFR equations tended to be more accurate compared to creatinine-based eGFR (30% accuracy 76.3–89.5% versus 76.3–78.9%, respectively), which led to significant overexposure, especially in younger (aged ≤ 2 years) children. Of all equations, the Schwartz cystatin C-based equation had the highest accuracy and lowest bias. Although anthropometric dosing performed comparably to many of the eGFR equations overall, we observed a weight-dependent change in bias leading to underdosing in the smallest patients. Using cystatin C-based eGFR equations for carboplatin dosing in children leads to more accurate carboplatin-exposure in patients with normal renal function compared to anthropometric dosing. In children with impaired kidney function, this trend might be more pronounced. Anthropometric dosing is hampered by a weight-dependent bias.

## 1. Introduction

Carboplatin is a second-generation platinum-containing compound commonly used in pediatric oncology, mainly for the treatment of solid tumors but also for the treatment of low-grade glioma and retinoblastoma [1,2]. Although generally not labelled for pediatric patients, carboplatin is frequently used in children, mostly in the treatment of solid tumors. This is due to the fact that carboplatin is associated with less renal and neurological toxicity than cisplatin [3]. For the treatment of solid tumors in adults, carboplatin dosing based on renal function is recommended since carboplatin is almost exclusively (up to 80%) eliminated by glomerular filtration. Its pharmacokinetics are therefore mainly determined by the patient’s renal function [4,5]. The linear relationship between carboplatin clearance and glomerular filtration rate (GFR) initiated the development of carboplatin dosing equations based on renal function in adults [4] as well as children [6,7]. Renal function-based dosing results in more reproducible and reliable drug exposure than anthropometric dosing [8,9] and real-time monitoring with adaptive dosing has been associated with more consistent platinum exposure in children [10]. The importance of controlling carboplatin exposure is illustrated by the relationship between the carboplatin area under the concentration-time curve (AUC; equals dose divided by drug clearance), toxicity [11,12], and clinical efficacy [13]. Renal function-based dosing is of particular importance in children as it corrects for changes in renal function during childhood [14], but has not yet been implemented in standard clinical practice. One of the most common types of toxicity is ototoxicity, which is particularly relevant in children with retinoblastoma, who may also suffer from impaired vision following treatment of their underlying illness [2].

The different dosing equations were developed using gold standard GFR measurements (i.e., ^51^Chromium ethylenediamine tetraacetic acid (^51^Cr-EDTA) clearance and technetium-99 m diethylenetriaminepentacetic acid (^99^mTc-DTPA) clearance). Although highly accurate, these methods are not widely available and often too invasive and time-consuming for routine measurement of GFR [15]. Therefore, current guidelines in clinical nephrology advise the use of estimated GFR (eGFR) based on the serum concentrations of creatinine or cystatin C [16,17]. Carboplatin dosing using eGFR may therefore be a more practical alternative to calculate the appropriate carboplatin dose. This has been studied extensively in adults [18,19].

In children, a number of eGFR equations have been developed using either creatinine and/or cystatin C [20]. These different eGFR equations might be used instead of a gold standard GFR measurement in the widely used Newell equation to calculate carboplatin clearance [6]. However, the only study so far evaluating eGFR for carboplatin dosing in pediatric oncology patients compared eGFR with a gold standard GFR measurement and extrapolated these findings to drug exposure [15].

The aim of the present study was to test the accuracy of several eGFR-based carboplatin dosing in children by the Newell equation using measured carboplatin exposure as the gold standard.

## 2. Materials and Methods

### 2.1. Patients and Treatment

To be eligible for study participation, patients had to be between 0 and 18 years old and scheduled to receive carboplatin for the treatment of retinoblastoma. Patients received an intravenous carboplatin dose of 560 mg/m^2^ (in children ≥10 kg) or 18.7 mg/kg (in children <10 kg) during one hour. Concomitant chemotherapeutic drugs were vincristine 1.5 mg/m^2^ and etoposide 150 mg/m^2^, administration of which are part of the local standard treatment protocol.

### 2.2. Ethical Approval

All procedures performed in studies involving human participants were in accordance with the ethical standards of the Institutional Review Board of Amsterdam UMC, location VUmc (formerly known as VUmc) and with the 1964 Helsinki declaration and its later amendments or comparable ethical standards. Written informed consent was signed by all patients or parents/guardians, as appropriate, before participating in the study.

### 2.3. Carboplatin Administration, Blood Sampling and Platinum Analysis

Carboplatin was administered intravenously for 60 min at a constant rate, diluted in 5% dextrose at doses ranging from 80 mg to 430 mg (median 340 mg) via central venous catheters. The drug was administered by study physicians or nurses, who punctually recorded dosing and sampling times in case report forms. After administration of carboplatin, the central venous line was flushed with 0.9% NaCl for 15 min and all connectors were changed to prevent contamination during blood sampling from the indwelling line. This procedure was evaluated ex vivo and showed no contamination of carboplatin originating from the indwelling line (manuscript in preparation). At various time points, venous blood (2 mL) was collected into tubes containing lithium heparin anticoagulant. Samples were obtained at 2.5, 8, 10, and 23 h after the start of infusion. These time points were determined using WinPOPT version 1.1 (Otago, New Zealand). An optimal design was calculated based on previously published carboplatin population pharmacokinetics [21]. To minimize the burden for outpatients, a limited sampling strategy with two samples (1.5 and 5 h after start of infusion) was applied in three patients in four studies. The choice of sampling points was based on population PK modeling and on sampling points described earlier in the literature for carboplatin [5,6,21]. These were deemed the optimal timepoints to cover the expected distribution–elimination curve. The concentration–time curves were visually inspected to look for goodness of fit and possibe bias.

After the sampling procedure, all samples were immediately placed in an ice bath and centrifuged (1500× *g* at 4 °C for 10 min), to separate plasma from whole blood. In order to separate total from protein-bound platinum, ethanol precipitation was used immediately after sample collection since this method requires less blood and is less time consuming than ultrafiltration. The ethanol precipitation method was validated to be similar to the ultrafiltration methodology for both cisplatin and carboplatin [22] and was therefore preferred for the current study. It has been used extensively by us in other studies for which both methods were compared and showed to be similar as well [23,24], as described previously immediately after sample collection [25]. An aliquot of plasma (0.5 mL) was thoroughly mixed with 1.5 mL of ice cold (−20 °C) ethanol and incubated for several hours at −20 °C to denaturize the plasma proteins. This mixture was centrifuged again at 4 °C and 1500× *g* for 10 min. The supernatant was carefully transferred to a 2.0 mL tube and stored at −20 °C until assay by flameless atomic absorption spectrophotometry (AA 800, Perkin Elmer, Waltham, MA, USA) [24,25]. The performance of this analytical procedure was comparable to the original assays [23,24,25], regarding linearity (linear in the range of the used calibration line of 2.5–40 µM carboplatin), lower limit of quantitation for total platinum (2.5 µM), within and between day accuracy of total platinum (12–13%) and unbound platinum (3%).

### 2.4. Renal Function

Renal function was assessed before each course of carboplatin. Creatinine was measured using an enzymatic method (Modular Analytics, Roche Diagnostics, Mannheim, Germany), which is traceable to isotope dilution mass spectrometry [26]. Cystatin C was measured from a frozen serum sample within one run using a particle-enhanced immunonephelometric assay (Siemens Healthcare, Marburg, Germany) on a Dade Behring Nephelometer II, which was traceable to the IFCC standard [27]. Estimated GFR was calculated using six equations, which were developed in pediatric populations [28]: the creatinine-based Schwartz_crea_ [29], Brandt [30] and Millisor [15] equations and the cystatin C-based Schwartz_cys_ [29], FAS_cys_ [31] and Berg [32] equations. Details of these equations can be found in Table 1. Choice of equations was based on performance in validation studies using gold standard GFR measurements [33]. Of note, both the Brandt and Millisor equations were derived in pediatric oncology patients.

### 2.5. Pharmacokinetic and Statistical Analysis

Our calculations were based on the population pharmacokinetic analysis for carboplatin as used in the study by Ekhart et al [34]. Briefly, a two-compartment model was developed with first-order elimination which describes carboplatin concentration–time profiles: clearance was 8.38 L/h; volume of distribution was 15.4 L and the distribution micro-constants k12 and k21 were 0.135 h^-1^ and 0.215 h^-1^, respectively. This model was implemented in MWPharm++ 1.35 (Mediware a.s., Praha, Czech Republic). The volume of distribution was allometrically scaled to corrected lean body mass. Pharmacokinetic parameters were assumed to be distributed log-normally. Residual error was assumed to be distributed normally with a standard deviation according to SD = 0.1 × C, in which C is platinum plasma concentration. Individual clearance was calculated for each patient and cycle using an iterative Bayesian procedure. The Marquardt method was used with a stop criterion of 1.00 × 10^−6^.

Measured carboplatin clearance obtained from these individual fits were compared with predicted carboplatin clearance using the Newell equation [6]:predicted carboplatin clearance (mL/min) = eGFR (mL/min) + 0.36 × body weight (kg)

As most eGFR equations yield standardized GFR in mL/min/1.73 m^2^ GFR was converted to absolute GFR in mL/min before use in the Newell formula. Body surface area (BSA) was calculated according to Mosteller [35]. Predicted and measured carboplatin clearance were used to calculate predicted and measured drug exposure by means of the area under the concentration time curve (AUC).
AUC (mg/mL.min) = dose (mg)/clearance (mL/min)

The main outcome parameters of this study were predicted carboplatin AUC compared to measured carboplatin AUC, i.e., carboplatin exposure.

Performance of the different eGFR equations in predicting carboplatin exposure was compared using the following parameters:The percentage prediction error (%PE), defined as: $\frac{\left(\mathrm{observed}\;\mathrm{AUC}-\mathrm{predicted}\;\mathrm{AUC}\right)}{\mathrm{observed}\;\mathrm{AUC}}\times 100\%$ which is a measure of bias;The absolute percentage prediction error (APE) $|\frac{\left(\mathrm{observed}\;\mathrm{AUC}-\mathrm{predicted}\;\mathrm{AUC}\right)}{\mathrm{observed}\;\mathrm{AUC}}|\times 100\%$, which is a measure of imprecision;Accuracy assessed by calculating the proportion of predicted AUC values within ± 30% of measured AUC (P_30_ accuracy), a commonly used accuracy measure in the evaluation of eGFR equations [20].

As limits of overdosing by more than 25% and underdosing by more than 10% appear to be clinically relevant in oncology patients [19], we also calculated the proportion of predictions between 125% and 90% of measured AUC (P_−10 to +25_) in analogy to the study by Millisor et al [15]. Since the local standard retinoblastoma treatment protocol used anthropometric dosing based on BSA for children over 10 kg and on body weight for smaller children to achieve a carboplatin exposure within the target AUC, this approach was evaluated by comparing target AUC and measured AUC using the parameters described above (%PE, APE and accuracy). The target AUC of carboplatin used in this study was 7.42 mg/mL.min and was based on previously published data by Newell et al [6]. 

Given the physiological differences in body dimensions and renal function [20] between young infants and older children, we analyzed patients under two years separately from older children. Furthermore, since there was a dosing difference in children below and above 10 kg, these were also analyzed separately.

Continuous variables are presented as median (interquartile range; IQR). Qualitative variables are displayed as numbers (%). Binary performance outcome of the different eGFR equations were compared using the McNemar test, Chi-squared test or Fisher’s exact test. Repeated measurements were considered as independent, therefore continuous measures of performance were tested using a Wilcoxon signed rank test. Diagnostic graphs and additional statistical analyses were made using IBM SPSS statistics 22 (Chicago, IL, USA) and GraphPad Prism 7.0 (San Diego, CA, USA). All tests were done at a two-sided significance level of 0.05.

## 3. Results

### 3.1. Patients

In total, 13 children were included. The patient characteristics are summarized in Table 2. Of all of the patients, 46.2% were female. All patients had normal renal function, defined as creatinine and cystatin C levels below the age-appropriate upper limit. In total, 38 clearance studies were performed, of which 18 (47%) were performed in children under the age of two years. All but two patients were studied during multiple courses (two during two courses, four during three courses and five during four courses).

### 3.2. Pharmacokinetics and Comparison of Equations

As expected [20] eGFR normalized for BSA was higher in children older than two years, while all patients had normal for age kidney function. Of note, the difference in eGFR between both age groups was highest for the Berg equation and lowest for the Schwartz_crea_ equation. Observed carboplatin clearance ranged from 11.68 mL/min to 57 mL/min (median 33.42 mL/min), and observed AUC ranged from 3.56 mg/mL.min to 12.81 mg/mL.min (median 7.29 mg/mL.min). After visual inspection of the concentration–time curves there was an adequate goodness of fit of the final model and no bias was observed. In line with the eGFR data, observed carboplatin clearance differed between both age groups even when normalized for BSA (84.5 mL/min/1.73 m^2^ in the younger versus 116.7 mL/min/1.73 m^2^ in the older patients).

The performance of all six eGFR-based equations as well as anthropometric dosing in predicting carboplatin AUC is presented in Table 3. All dosing methods resulted in a positive bias with median %PE values ranging from 4.1 to 21%. In the total group as well as the two subgroups, the Schwartz_cys_ equation performed best both in terms of bias and accuracy. In the infant group, Schwartz_cys_ and FAS_cys_ had strikingly high accuracy, while the Brandt equation, which was developed specifically for young children, did not outperform the other creatinine-based equations. In the older children, Schwartz_crea_, Schwartz_cys_ and Millisor had the least bias and highest precision and yielded high P_30_ accuracy. Combining a cystatin C- and a creatinine-based eGFR yielded similar results. Of note, anthropometric dosing had lower %PE, APE and higher accuracy when compared to a number of the eGFR equations.

In Figure 1, %PE of the different eGFR equations and anthropometric dosing in the individual studies is shown as waterfall plots. For all methods, the plot is skewed towards positive %PE indicating a trend towards overdosing of carboplatin. This is more marked for the creatinine-based than the cystatin C-based eGFR equations and anthropometric dosing. Overall, the extremes of both positive and negative accuracy were less in the cystatin C-based equations. Dosing based on Schwartz_cys_ resulted in the flattest waterfall plot indicating that this is the most balanced method. In Figure 2, %PE in anthropometric dosing appears to be directly related to body weight, leading to significant underdosing in the youngest children, while %PE is mostly positive in studies above 10 kg. This was not observed with any of the eGFR-based dosing methods.

The dotted lines indicate the thresholds of +25% and −10%, also used for the calculation of accuracy (−10 to 25%).

Each symbol indicates the same patients in all figures. The dotted lines indicate the thresholds of percentage error of +25% and −10%.

## 4. Discussion

The present study is the first external evaluation of carboplatin dosing using estimated rather than measured GFR in the pediatric Newell equation. In our small series of children with retinoblastoma with normal kidney function, cystatin C-based carboplatin dosing using the Schwartz_cys_ equation led to more accurate carboplatin exposure than the other cystatin C- or creatinine-based eGFR methods. While anthropometric dosing based on BSA or weight performed reasonably well overall, we observed significant underdosing of carboplatin in studies performed at a bodyweight below 10 kg, although patient numbers were small.

When GFR is measured using a gold standard technique, a strong relationship with carboplatin clearance has been demonstrated [6]. However, gold standard GFR measurements using inulin, ^51^Cr-EDTA, ^99^Tc-DTPA, ^125^I-iothalamate or iohexol [14,36,37] are invasive, costly and cumbersome, thereby precluding widespread use in clinical practice. Therefore, current international guidelines advocate the use of eGFR based on endogenous markers for drug dosing [38] and the detection, evaluation, and management of kidney disease [16,17]. This has led to the development of a wide range of pediatric eGFR equations based on creatinine and cystatin C in recent years [20].

Originating from muscle metabolism, serum creatinine not only reflects GFR but is also influenced by extrarenal factors, such as age, gender, and muscle mass. In children, muscle mass increases with age so that serum creatinine can only be used to monitor renal function after correction for anthropometric data or age [39]. This leads to a particular shortcoming in pediatric oncology patients, where muscle wasting from treatment and malignancy may cause inappropriately low creatinine values thereby masking kidney failure. This can result in suboptimal treatment [40] since underestimation of creatinine will lead to an overestimation of GFR and overdosing of carboplatin with the potential of increased toxicity. These shortcomings of creatinine in pediatric oncology patients have been demonstrated by Blufpand et al. [41]. Therefore, Brandt et al. [30] and Millisor et al. [15] developed specific creatinine-based eGFR equations for children with a malignancy. In our study, however, neither of these two equations performed better than the general Schwartz_crea_ equation and had higher %PE values than Schwartz_cys_, indicating overestimation of GFR. The Brandt equation had the highest positive bias of all creatinine-based equations. This was also observed in Millisor’s paper, where the Brandt equation systematically overestimated GFR [15]. The cystatin C-based equations performed slightly better than the creatinine-based equations, in particular in younger children. This is remarkable as both Schwartz_cys_ and FAS_cys_ were developed in children above the age of one [29,42], while the youngest patient in the cohort of Brandt et al. was 2.8 months of age [30] and 1 month in Millisor’s cohort [15]. 

In unselected populations, creatinine-based eGFR equations perform comparably to cystatin C-based equations and the arithmetic mean of the two yields the best results [28,33]. A weighted mean of 40% eGFR_creat_ and 60% eGFR_cys_ has been suggested to be optimal for patients with malignancy [43]. Still, this did not improve accuracy of carboplatin exposure in the present study because all equations had a positive bias. 

Although largely eliminated renally, roughly 20% of carboplatin is non-renally cleared, which to a large extent is due to tissue binding, with a small proportion excreted through the liver [11,44]. Calvert et al. found a relatively stable extra-renal clearance of carboplatin equivalent to about 25 mL/min in adults [4]. This was adjusted for children by introducing a weight-dependent term of 0.36 (i.e., 25 divided by the weight of an average adult of 70 kg) to be added to GFR in the Newell equation [6]. This approach assumes that the extra-renal clearance of carboplatin is linearly related to weight. Based on our data this approach seems appropriate as we observed no systematic relation between %PE and bodyweight in the Newell equation with any of the eGFR methods used. 

By contrast, the anthropometric dosing regimen based on weight in small children and BSA in children above 10 kg was associated with underdosing in the former. This is not unexpected as total body water for weight [45] as well as BSA for weight (infants versus older children 0.046 vs. 0.042 m^2^/kg in our cohort) are inversely related to age. Therefore, weight-based dosing will result in lower carboplatin exposure in small children, as observed here.

It should be borne in mind that the Calvert equation and, by extension, the Newell equation, are based on measured GFR in mL/min. As illustrated by Beumer et al. [46] in their commentary on a recently published new model for GFR estimation in adults [47], each conversion step, i.e., from measured GFR to estimated GFR [20] and also from GFR in mL/min to GFR in mL/min/1.73 m^2^ introduces the potential for bias. It would therefore be useful to recalibrate the Newell equation in children using eGFR instead of measured GFR and also re-evaluate the extra-renal clearance of carboplatin. This could be a most welcome follow-up study.

We acknowledge a number of limitations in our study. First, we did not use a gold standard GFR technique for direct comparison with the eGFR equations and did not assess the most accurate CKiD-3 equation incorporating anthropometric data, creatinine, urea and cystatin C [29,43], as urea had not been measured. However, although the CKiD-3 equation is very accurate for eGFR measurements, it is not feasible in this population since carboplatin administration is always combined with hyperhydration, which strongly affects urea concentrations [48]. Therefore, it is assumed there is limited additional value of adding urea in an equation for this specific population. Furthermore, accuracy of carboplatin exposure in the present study compares surprisingly well to published data comparing eGFR equations with gold standard GFR measurements as reported in pediatric oncology studies [15,49]. P_30_ accuracy of carboplatin exposure prediction using FAS_cys_ and Schwartz_cys_ was well above the 80% benchmark used for eGFR equations [33]. Second, the GFR in our patients was normal, which may have reduced the potential benefit of kidney function-based dosing when compared to anthropometric dosing. In a study in adults, Ekhart et al. [50] found no advantage of eGFR-based dosing in patients with a GFR above 50 mL/min. Veal et al. [10] demonstrated that GFR-based dosing can lead to overdosing in children with hyperfiltration, a not uncommon finding in patients with malignancy undergoing hyperhydration [51]. Still, none of the patients here showed hyperfiltration as all eGFR measurements were below 140 mL/min/1.73 m^2^.

Third, the age spectrum of our study population differed from the populations in which some of the eGFR equations were calibrated. We specifically chose the Brandt and Millisor equations since they were derived in populations including very young children, while the Schwartz equations [29] and the FAS equations [31] were derived in children older than one year. Of note, only four infants participated in the development of the Newell equation [6]. Despite these population differences, we found an astonishingly good performance of the different equations in children aged two years or younger.

Fourth, calculation of the observed carboplatin AUC was based on the publication by Ekhart et al. [34], who did not report bias or imprecision of their method, and visual inspection of the goodness of fit of the final model. These characteristics of the standard intrinsically influence the outcome of comparison of the different dosing methods.

Finally, the sample size of this study was small and included repeated measurements in most patients. We only included retinoblastoma patients resulting in a very homogenous and young population without much comorbidity. Therefore, it remains to be demonstrated whether our results can be extrapolated to other populations, in particular to patients with a more diverse range of kidney function, but it is conceivable that differences will be more pronounced in this group of patients. Our results need to be confirmed in a larger cohort, including a larger age spectrum, to determine its clinical relevance.

## 5. Conclusions

In conclusion, cystatin C-based eGFR equations for carboplatin dosing in children lead to more accurate carboplatin exposure. While performing well in the overall analysis, anthropometric dosing is hampered by a weight-dependent change in bias.

## Figures and Tables

**Figure 1 cancers-13-05963-f001:**
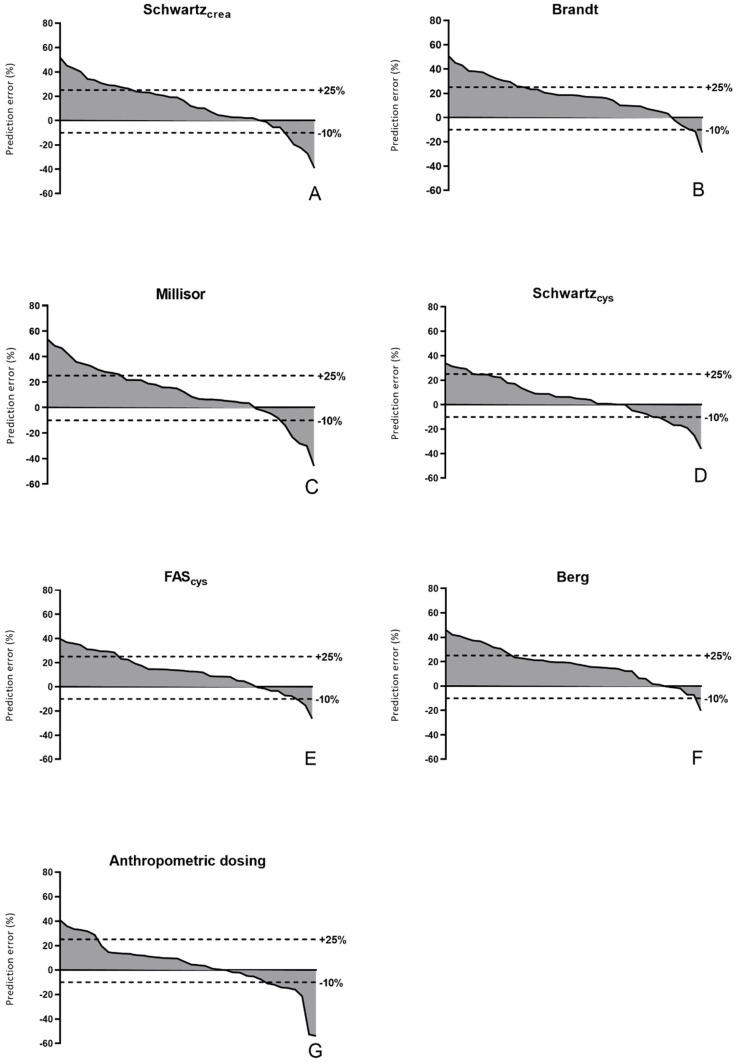
Percentage prediction error (%PE) of the different dosing methods.

**Figure 2 cancers-13-05963-f002:**
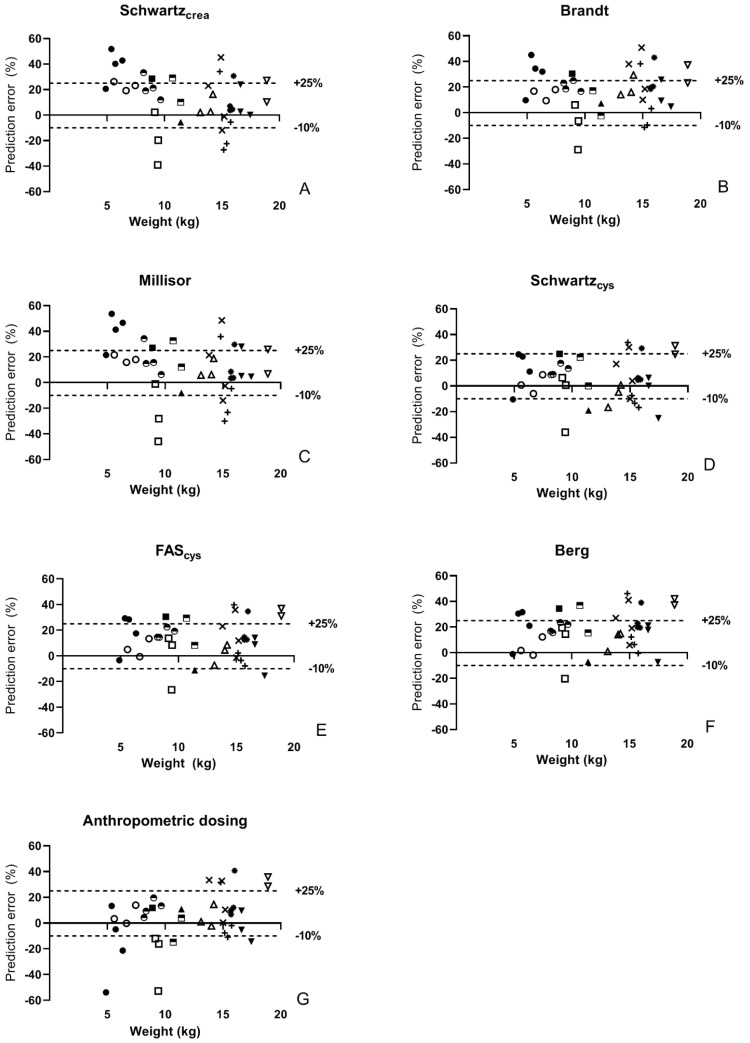
Percentage prediction error according to weight in all patterns.

**Table 1 cancers-13-05963-t001:** Equations used for calculation of estimated GFR.

**Equations Based on Creatinine**	
Equation 1	$eGFR-Schwartz\;creatinine\;(\mathrm{mL}/\min/1.73\;m^{2})=42.3\times {\left(\frac{height\;\left(m\right)}{creatinine\;\left(\mathrm{mg}/\mathrm{dL}\right)}\right)}^{0.79}$
Equation 2	$eGFR-Brandt\;\left(\frac{\mathrm{mL}}{\min}\right)=k\times \sqrt{\frac{\left(age\;\left(months\right)+6\right)\times \left(weight\left(\mathrm{kg}\right)\right)}{creatinine\left(\frac{\mathrm{mg}}{\mathrm{dL}}\right)}}\;$
	k=0.95 (females) and 1.05 (males)
Equation 3	$eGFR-Millisor\;\left(\mathrm{mL}/\min/1.73\;m^{2}\right)=0.33\times \left(\frac{height\;\left(\mathrm{cm}\right)}{creatinine\;\left(\mathrm{mg}/\mathrm{dL}\right)}\right)$
**Equations Based on Cystatin C**	
Equation 4	$eGFR-Schwartz\;cystatin\;C\;\left(\mathrm{mL}/\min/1.73\;m^{2}\right)=40.6\times {\left(\frac{1.8}{cystatin\;C\;(\mathrm{mg}/L}\right)}^{0.93}$
Equation 5	$eGFR-FAS\;cystatin\;C\;\left(\mathrm{mL}/\min/1.73\;m^{2}\right)=107.3\;÷\;\left(\frac{cystatin\;C\;\left(\mathrm{mg}/L\right)}{0.82}\right)$
Equation 6	$eGFR-Berg=91\times cystatin\;C\;{\left(\mathrm{mg}/L\right)}^{-1.213}$

eGFR: estimated glomerular filtration rate.

**Table 2 cancers-13-05963-t002:** Patient characteristics.

Number of Clearance Studies	Total 38	<2 Years 18	>2 Years 20	*p*-Value
Age, years	2.2 [0.5–3.3]	0.5 [0.4–1.0]	3.3 [3.0–4.0]	<0.001
Body weight, kg	13.5 [8.8–15.7]	8.6 [6.2–9.5]	15.6 [14.8–16.5]	<0.001
BSA, m^2^	0.60 [0.41–0.66]	0.40 [0.33–0.43]	0.65 [0.63–0.67]	<0.001
BMI, kg/m^2^	15.9 [15.1–18.6]	17.9 [14.7–18.9]	15.6 [15.1–16.4]	0.11
Creatinine, mg/dL ^a^	0.27 [0.22–0.32]	0.25 [0.18–0.29]	0.31 [0.24–0.33]	0.008
Cystatin C, mg/L	0.80 [0.74–0.97]	0.97 [0.86–1.09]	0.75 [0.69–0.79]	<0.001
eGFR-Schwartz_crea_ (mL/min/1.73 m^2^)	104.3 [92.8–120.7]	95.2 [82.6–116.7]	108.9 [99.9–125.3]	<0.001
eGFR-Schwartz_crea_ abs (mL/min)	36.0 [21.7–43.0]	21.7 [19.1–26.6]	42.7 [39.3–44.9]	<0.001
eGFR-Brandt (mL/min/1.73 m^2^)	115.4 [97.9–128.7]	97.8 [85.2–106.0]	128.6 [123.4–138.9]	<0.001
eGFR-Brandt abs (mL/min)	43.5 [22.1–49.4]	22.0 [18.6–24.0]	49.2 [46.9–52.1]	<0.001
eGFR-Millisor (mL/min/1.73 m^2^)	103.4 [89.2–124.5]	92.2 [76.9–119.3]	109.2 [98.0–130.5]	<0.001
eGFR-Millisor abs (mL/min)	35.8 [21.7–43.3]	21.3 [18.2–26.9]	42.6 [37.8–46.4]	<0.001
eGFR-Schwartz_cys_ (mL/min/1.73 m^2^)	99.7 [84.1–107.4]	83.8 [75.3–94.2]	106.6 [100.9–114.7]	<0.001
eGFR-Schwartz_cys_ abs (mL/min)	33.6 [18.1–40.9]	18.0 [13.8–24.4]	40.6 [38.0–44.0]	<0.001
eGFR-FAScys (mL/min/1.73 m^2^)	109.4 [91.1–118.5]	90.7 [80.8–102.9]	117.6 [110.8–127.1]	<0.001
eGFR-FAScys abs (mL/min)	37.2 [19.6–45.2]	19.4 [14.7–26.7]	44.8 [41.7–48.7]	<0.001
eGFR-Berg (mL/min/1.73 m^2^)	118.5 [94.9–130.6]	94.5 [82.1–110.1]	129.3 [120.3–142.2]	<0.001
eGFR-Berg abs (mL/min)	41.3 [20.2–49.9]	20.1 [14.8–28.5]	49.2 [45.0–54.6]	<0.001
Observed carboplatin clearance (mL/min/1.73 m^2^)	104.3 [83.6–122.9]	84.5 [75.3–106.2]	116.7 [90.3–136.1]	0.002
Observed carboplatin clearance abs (mL/min)	33.4 [18.8–45.4]	18.7 [15.9–27.9]	44.4 [37.3–49.3]	<0.001
Observed carboplatin AUC (mg/mL.min)	7.9 [7.0–8.6]	7.7 [6.4–8.5]	8.2 [7.3–10.7]	0.09

^a^ To convert to µmol/L multiply by 88.4. BMI: body mass index, BSA: body surface area, eGFR: estimated glomerular filtration rate.

**Table 3 cancers-13-05963-t003:** Performance of predicted carboplatin AUC values based on different estimates of GFR and anthropometric dosing. Data for infants (<2 years) and older children are displayed separately.

		Bias (mg/mL.min)	%PE (%)	APE (%)	Accuracy (±30%)	Accuracy (−10 to 25%)
Schwartz_crea_	Total *N* = 38	1.1 [0.1 to 2.5]	14.2 [1.7 to 27.6]	20.1 [5.5 to 28.8]	78.9	57.9
Brandt		1.5 [0.6 to 2.4]	18.3 [8.8 to 29.7]	18.5 [9.7 to 29.7]	76.3	63.2
Millisor		1.0 [0.2 to 2.3]	13.6 [2.3 to 27.4]	18.4 [6.3 to 29.8]	76.3	57.9
Schwartz_cys_		0.4 [−0.5 to 1.6]	5.7 [−6.3 to 18.9]	10.9 [5.7 to 23.4]	89.5	65.8
FAScys		1.1 [0.1 to 2.0]	13.1 [1.6 to 24.5]	13.9 [8.1 to 26.9]	84.2	68.4
Berg		1.3 [0.4 to 2.3]	18.5 [6.3 to 27.9]	19.3 [11.0 to 27.9]	76.3	71.1
Anthropometric dosing		0.4 [−0.4 to 1.2]	5.6 [−5.9 to 13.6]	12.0 [5.2 to 20.1]	81.6	55.3
Schwartz_crea_	Infants *N* = 18	1.7 [0.6 to 2.5]	21.0 [8.2 to 30.2]	22.3 [17.4 to 34.8]	72.2	50.0
Brandt		1.4 [0.5 to 2.0]	17.0 [6.9 to 26.4]	17.6 [8.9 to 29.2]	77.8	66.7
Millisor		1.3 [0.4 to 2.4]	16.9 [4.5 to 33.0]	21.6 [14.3 to 36.1]	66.7	55.6
Schwartz_cys_		0.7 [−0.1 to 1.5]	8.8 [−1.5 to 18.9]	10.9 [6.4 to 22.5]	94.4	77.8
FAS_cys_		1.1 [0.3 to 1.9]	14.1 [3.7 to 24.0]	14.6 [8.4 to 26.9]	94.4	66.7
Berg		1.3 [0.1 to 2.2]	16.3 [1.0 to 25.3]	18.2 [11.0 to 25.3]	77.8	72.2
Anthropometric dosing		0.3 [−1.0 to 1.0]	3.6 [−15.1 to 12.2]	12.7 [4.7 to 17.0]	88.9	55.6
Schwartz_crea_	Older children *N* = 20	0.3 [−0.1 to 2.4]	4.1 [−0.7 to 23.7]	11.1 [3.0 to 26.3]	85.0	65.0
Brandt		1.5 [0.7 to 3.7]	19.0 [9.5 to 35.3]	19.0 [10.4 to 35.3]	75.0	60.0
Millisor		0.4 [−0.1 to 2.4]	6.1 [−1.1 to 24.8]	11.3 [4.8 to 27.5]	85.0	60.0
Schwartz_cys_		0.4 [−0.7 to 2.4]	4.3 [−9.4 to 22.9]	11.7 [4.8 to 25.0]	85.0	55.0
FAS_cys_		1.0 [−0.0 to 3.1]	12.3 [−0.4 to 29.2]	12.7 [7.3 to 29.2]	75.0	70.0
Berg		1.6 [0.5 to 3.7]	19.3 [7.9 to 34.7]	19.3 [8.6 to 34.7]	75.0	70.0
Anthropometric dosing		0.8 [−0.2 to 3.3]	9.7 [−2.1 to 30.9]	10.7 [5.7 to 30.9]	75.0	55.0

APE: absolute percentage prediction error, %PE: percentage prediction error.

## Data Availability

Data can be obtained at reasonable request.

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
