# Peer review of "Carboplatin Dosing in Children Using Estimated Glomerular Filtration Rate: Equation Matters"

_cancers, 2021, doi:10.3390/cancers13235963_

Round 1

Reviewer 1 Report

The study of van de Velde et al. is an interesting study validating a carboplatin dosing method based on kidney function. Carboplatin dosing based on cystatin C resulted in more adequate dosing than dosing on body surface area or weight. The major limitations are listed. However, it is remarkable that the most accurate CKiD-3 equation could not be measured based on the fact that urea was not measured. Were these blood samples not saved so that this routine parameter could still be measured?

The cystatin C-based equations performed slightly better than the creatinine-based equations, in particular in younger children. This is remarkable as both Schwartzcys and FAScys were developed in children above the age of one. What could be the underlying reasons of this finding?

Author Response

Reviewer #1

General comments

The study of van de Velde et al. is an interesting study validating a carboplatin dosing method based on kidney function. Carboplatin dosing based on cystatin C resulted in more adequate dosing than dosing on body surface area or weight. The major limitations are listed.

Specific comments

Comment 1

However, it is remarkable that the most accurate CKiD-3 equation could not be measured based on the fact that urea was not measured. Were these blood samples not saved so that this routine parameter could still be measured?

Reply comment 1
Unfortunately, urea was not measured in this study because serum urea concentrations are strongly affected by hyperhydration, which is part of the supportive treatment around carboplatin administration. Even though hypothetically the CKiD-3 equation, which combines creatinine, urea and cystatin C, would be an interesting equation for dosing carboplatin, it is not feasible in this specific population due to changing urea clearance depending on hydration status [1]. We acknowledge that it is important to address this. Therefore, we have added in the Discussion part of our manuscript (lines 329-333):
“However, although the CKiD-3 equation is very accurate for eGFR measurements, it is not feasible in this population since carboplatin administration is always combined with hyperhydration, which strongly affects urea concentrations. Therefore, it is assumed there is limited additional value of adding urea in an equation for this specific population.”

  1. den Bakker, E.G., R.J.B.J.; Bökenkamp, A.; . Endogenous markers for kidney function in children: a review. Critical Reviews in Clinical Laboratory Sciences 2018, 55, 163-183.

Comment 2
The cystatin C-based equations performed slightly better than the creatinine-based equations, in particular in younger children. This is remarkable as both Schwartzcys and FAScys were developed in children above the age of one. What could be the underlying reasons of this finding?

Reply Comment 2
We agree that the good performance of the two cystatin C based equations in young children is remarkable. However, it must be kept in mind that both cystatin C-based eGFR equations (i.e. Schwartzcys and FAScys) were developed in older children because there was insufficient clearance data in very young children to be included in the development cohorts [2]. Still, the relationship between cystatin C (unlike creatinine) and GFR is stable across all age-groups, which might explain our observation. Of course, this hypothesis needs to be confirmed using gold-standard clearance measurements in babies.

  1. Bokenkamp, A.; Domanetzki, M.; Zinck, R.; Schumann, G.; Brodehl, J. Reference values for cystatin C serum concentrations in children. Pediatr Nephrol 1998, 12, 125-129, doi:10.1007/s004670050419.

Reviewer 2 Report

The study presented is interesting. However, I have a few suggestions to clarify some aspects of the work.

Check and follow the recommendations Transparent reporting of a multivariable prediction model for individual prognosis or diagnosis (TRIPOD): The TRIPOD statement https://www.equator-network.org/reporting-guidelines/tripod-statement/
or equivalent.

Estimate the number of patients needed to be included to have sufficient power to test your hypothesis with adequate power.

Present the analytical and performance details of the platinum analytical technique.
Present the details of your population pharmacokinetic model (IIV, co-variance matrix, error model).
Has this model been externally validated? in patients of the same age? Same profile?

On what basis did you choose the times of biological sampling?

Figure 1: What do you represent on the X-axis?

Author Response

Reviewer #2

General Comments

The study presented is interesting. However, I have a few suggestions to clarify some aspects of the work.

Specific Comments

Comment 1
Check and follow the recommendations Transparent reporting of a multivariable prediction model for individual prognosis or diagnosis (TRIPOD): The TRIPOD statement https://www.equator-network.org/reporting-guidelines/tripod-statement/
or equivalent.
Estimate the number of patients needed to be included to have sufficient power to test your hypothesis with adequate power.

Reply comment 1
Thank you for addressing these recommendations to us. We are well aware that the number of patients in this study is limited. These results should first be confirmed in a larger cohort, with a power analysis analogous to the TRIPOD statement as mentioned, before it can be applied clinically. To emphasize this, we have added to the Discussion part of our manuscript (lines 359-361): “Our results need to be confirmed in a larger cohort, including a larger age spectrum, to determine its clinical relevance.”

Comment 2

Present the analytical and performance details of the platinum analytical technique.

Reply comment 2

We understand the additional value of reporting this information in the manuscript. Overall, the performance of the analytical procedure was comparable to the original assays as referred to in the manuscript (references 22-24). We have added to the Method section of the manuscript (lines 129-133) “The performance of this analytical procedure was comparable to the original assays, regarding linearity (linear in the range of the used calibration line of 2.5-40 µM carboplatin), lower limit of quantitation for total platinum (2.5 µM), within and between day accuracy of total platinum (12-13%) and unbound platinum (3%)”.

Comment 3

Present the details of your population pharmacokinetic model (IIV, co-variance matrix, error model).
Has this model been externally validated? in patients of the same age? Same profile?

Reply comment 3

As mentioned in the Method section in Lines 153-154, the population pharmacokinetic analysis for carboplatin was previously published and our model was similar to this publication [3]. More details regarding this pharmacokinetic model are mentioned in Lines 154-164. Additionally to this:       

IIV was expressed as SD, as shown in the Figure below

  1. Ekhart, C.; Rodenhuis, S.; Schellens, J.H.; Beijnen, J.H.; Huitema, A.D. Carboplatin dosing in overweight and obese patients with normal renal function, does weight matter? Cancer Chemother Pharmacol 2009, 64, 115-122, doi:10.1007/s00280-008-0856-x.

Comment 4
On what basis did you choose the times of biological sampling?

Reply comment 4

With WinPOPT version 1.1 (Otago, New Zealand, 2006) an optimal design was calculated using carboplatin population pharmacokinetics as previously published [4]. We agree that this is useful additional information, therefore we have added to the Method section or our manuscript (lines 112-114): These time points were determined using WinPOPT version 1.1 (Otago, New Zealand). An optimal design was calculated based on previously published carboplatin population pharmacokinetics.”

  1. Riccardi, R.; Riccardi, A.; Lasorella, A.; Di Rocco, C.; Carelli, G.; Tornesello, A.; Servidei, T.; Iavarone, A.; Mastrangelo, R. Clinical pharmacokinetics of carboplatin in children. Cancer Chemother Pharmacol 1994, 33, 477-483, doi:10.1007/BF00686504.

Comment 5

Figure 1: What do you represent on the X-axis?

Reply comment 5

Figure 1 is a waterfall plot, which represents on the X-axis the individual measurements of percentage prediction error based on the above mentioned equations and sorted from high to low. For graphical purposes, it was decided not to show the individual bars, but just the area below the measured values. We understand however how this can be confusing. Therefore, we have added to the footnotes of Figure 1 (lines 254-255): “Individual observations of %PE (shown on the Y-axis) according to each equation. Measurements were sorted on the X-axis from high %PE to low %PE.”.

Reviewer 3 Report

I pediatric oncology, doses of carboplatin are still individualised based on body weight or body surface area because stuies like yours are laking. Even if the conclusions are limited due to classical reason in pediatric studies, this is a great job about a real clinical question.

I just have 3 comments about the presentation and interpretation of the results :

  1. I understand that all carboplatin courses have been considered as independent. As most of patients received multiple infusion this should be precized in the method section.
  2. An estimation of intraindividual variability in actual carboplatin CL should be presented in the results. In figure 2 we can see that for some patients there is an intraindividual variability in percentage of prediction error. This point deserves to be explained ; is it due to an intraindividual variability of actual carboplatin clearance without varibility of predicted clearance because no chnage in patients’ characteristics, or is it due to a variability of predicted clearance because of no change in patients’ characteristics without change in actual CL ?
  3. In your discussion and conclusion, you claim that anthropometric method lead to underdosing for children <10 kg. This should be weighted by the fact that only two patients were underdosed at 2 and 3 occasions. One of they was well dosed at other occasion an the other was underdosed for at least one occasion by all equations.

Author Response

Reviewer #3

General comments
In pediatric oncology, doses of carboplatin are still individualised based on body weight or body surface area because studies like yours are lacking. Even if the conclusions are limited due to classical reason in pediatric studies, this is a great job about a real clinical question.

Reply general comment

We appreciate this positive comment.

Specific comments
Comment 1
I understand that all carboplatin courses have been considered as independent. As most of patients received multiple infusion this should be precised in the method section.

Reply comment 1

We agree with the reviewer that within our study, we used repeated measurements within individuals and ideally these data should have been analyzed using a technique correcting for dependent measurements, such as mixed model analyses or generalized estimating equation (GEE). To emphasize our chosen method of analysis, we have added to the Method section of our manuscript (line 202): “Repeated measurements were considered as independent, therefore continuous measures of performance were tested using a Wilcoxon signed rank test”. 

Comment 2
An estimation of intraindividual variability in actual carboplatin CL should be presented in the results. In figure 2 we can see that for some patients there is an intraindividual variability in percentage of prediction error. This point deserves to be explained ; is it due to an intraindividual variability of actual carboplatin clearance without varibility of predicted clearance because no chnage in patients’ characteristics, or is it due to a variability of predicted clearance because of no change in patients’ characteristics without change in actual CL?

Reply comment 2

The reviewer asks if it is possible to distinguish the contribution of intra-individual variability of the actual carboplatin clearance from the variability of the predicted clearance. In order to do so, we would need a third “Gold standard” for comparison which unfortunately does not exist. However, visual examination comparing %PE in individual patients across different dosing methods provides some insights to this matter. As actual carboplatin clearance is the same across the different dosing methods, the pattern of the individual %PE reflects variability in predicted clearance. From this, it is evident, that the patterns of the cystatin C-based methods differ from the creatinine-based and from the anthropometry-based methods. This highlights the contribution of intra-individual variability of the predicted clearance.

Comment 3
In your discussion and conclusion, you claim that anthropometric method lead to underdosing for children <10 kg. This should be weighted by the fact that only two patients were underdosed at 2 and 3 occasions. One of they was well dosed at other occasion and the other was underdosed for at least one occasion by all equations.

Reply comment 3

We agree with the reviewer that sample size was a limitation in this study. Therefore, our results should be replicated in a larger cohort to make any firm conclusions. In our view, this study serves as a starting point to search for alternative dosing methods for carboplatin in children. By no means we imply that based on this study alone, the best carboplatin dosing strategy in children was identified. However, it generates evidence that there is a problem with anthropometric carboplatin dosing and that alternative dosing strategies should be developed. To emphasize this limitation, we have added to the Discussion part of our manuscript (lines 266-267) “…although patient numbers were small” and (lines 359-361) “Our results need to be replicated in a larger cohort, including a larger age spectrum, to determine its clinical relevance”.

Round 2

Reviewer 2 Report

The authors responded to several of my comments. 

Has your poppk model used for the estimation of AUC been validated (internal, external validation)? Discuss the impact of biases on your main result!

Discuss the imprecision induced in the AUC calculation by your choice of time and number of samples. 

Author Response

Comment

  • Has your population pk model used for the estimation of AUC been validated (internal, external validation)? Discuss the impact of biases on your main result!
  • Discuss the imprecision induced in the AUC calculation by your choice of time and number of samples.

Reply comment

The population PK model used in this paper, as first published by Ekhart et al [1], was not validated in our study. However, this model was only used to fit the concentration-time data. These concentration-time curves were adequately fitted and no bias was observed by visual inspection.

Furthermore, the choice of sampling time-points was based on the experience obtained by us and other population PK studies investigating cisplatin and carboplatin pharmacokinetics, both in adults and in children (see e.g. Giacconi et al, Chatelut et al, Newell et al and Riccardi et al. [2-5]). The time span and distribution of sampling points of the current paper was comparable to these previously published studies. Although, more extensive carboplatin sampling over a larger time span might have improved precision of the concentration-time data and therefore AUC, this had to be weighed against the invasiveness of these measurements in a vulnerable population (children) and the possible complications (i.e. blood loss and risk of infection). Therefore, this limited sampling strategy in combination with the previously published data was deemed to be reliable and sufficient for adequate AUC measurement. This is supported by that the fact that neither in the study by Ekhart et al. [1], nor in the present study bias or imprecision of the observed carboplatin-AUC was determined. Although this limits the value of the observed carboplatin-AUC as standard in our study, still all the dosing methods were compared to the same standard and differences in imprecision can therefore be attributed to the respective method. Bias of the observed AUC, however, would have influenced our results on bias and accuracy of the dosing methods. To further emphasize this, we have added to the paper in the Method section (lines 116-120)
“The choice of sampling points was based on population PK modeling and on sampling points described earlier in the literature for carboplatin [5,6,21]. These were deemed the optimal time-points to cover the expected distribution-elimination curve. The concentration-time curves were visually inspected to look for goodness of fit and possible bias”
and in the Results section (lines 228-229):
“After visual inspection of the concentration time-curves there was an adequate goodness of fit of the final model and no bias was observed.”
Finally, to discuss this method carboplatin AUC measurement, it was added to the Discussion part of the paper (lines 359-362):
Fourth, calculation of the observed carboplatin-AUC was based on the publication by Ekhart et al.[34] who did not report bias or imprecision of their method, and visual inspection of the goodness of fit of the final model. These characteristics of the standard intrinsically influence the outcome of comparison of the different dosing methods.”

  1. Ekhart, C.; Rodenhuis, S.; Schellens, J.H.; Beijnen, J.H.; Huitema, A.D. Carboplatin dosing in overweight and obese patients with normal renal function, does weight matter? Cancer Chemother Pharmacol 2009, 64, 115-122, doi:10.1007/s00280-008-0856-x.
  2. Giaccone, G.; Gonzalez-Larriba, J.L.; van Oosterom, A.T.; Alfonso, R.; Smit, E.F.; Martens, M.; Peters, G.J.; van der Vijgh, W.J.; Smith, R.; Averbuch, S., et al. Combination therapy with gefitinib, an epidermal growth factor receptor tyrosine kinase inhibitor, gemcitabine and cisplatin in patients with advanced solid tumors. Ann Oncol 2004, 15, 831-838.
  3. Chatelut, E.; Boddy, A.V.; Peng, B.; Rubie, H.; Lavit, M.; Dezeuze, A.; Pearson, A.D.; Roche, H.; Robert, A.; Newell, D.R., et al. Population pharmacokinetics of carboplatin in children. Clin Pharmacol Ther 1996, 59, 436-443, doi:10.1016/s0009-9236(96)90113-7.
  4. Newell, D.R.; Pearson, A.D.; Balmanno, K.; Price, L.; Wyllie, R.A.; Keir, M.; Calvert, A.H.; Lewis, I.J.; Pinkerton, C.R.; Stevens, M.C. Carboplatin pharmacokinetics in children: the development of a pediatric dosing formula. The United Kingdom Children's Cancer Study Group. J Clin Oncol 1993, 11, 2314-2323, doi:10.1200/jco.1993.11.12.2314.
  5. Riccardi, R.; Riccardi, A.; Lasorella, A.; Di Rocco, C.; Carelli, G.; Tornesello, A.; Servidei, T.; Iavarone, A.; Mastrangelo, R. Clinical pharmacokinetics of carboplatin in children. Cancer Chemother Pharmacol 1994, 33, 477-483, doi:10.1007/BF00686504.